**Data Availability Statement:** All relevant data are within the paper and its Supporting Information files.

# Incidence and predictors of preterm neonatal mortality at Mbarara Regional Referral Hospital in South Western Uganda

**Leevan Tibaijuka**[1,2]*, **Stephen M. Bawakanya**[1,2], **Asiphas Owaraganise**[1,3], **Lydia Kyasimire**[4], **Elias Kumbakumba**[4], **Adeline A. Boatin**[5], **Musa Kayondo**[1,2], **Joseph Ngonzi**[1,2], **Stephen B. Asiimwe**[6], **Godfrey R. Mugyenyi**[1,2]

1 Faculty of Medicine, Mbarara University of Science and Technology, Mbarara, Uganda, 2 Department of Obstetrics and Gynaecology, Mbarara Regional Referral Hospital, Mbarara, Uganda, 3 Infectious Diseases Research Collaboration, Kampala, Uganda, 4 Department of Paediatrics and Child Health, Mbarara University of Science and Technology, Mbarara, Uganda, 5 Department of Obstetrics and Gynaecology, Massachusetts General Hospital, Harvard Medical School, Boston, MA, United States of America, 6 Department of Epidemiology and Biostatistics, University of California San Francisco, California, San Francisco, United States of America

* leevantibs@gmail.com

## Abstract

### Introduction

Preterm neonatal mortality contributes substantially to the high neonatal mortality globally. In Uganda, preterm neonatal mortality accounts for 31% of all neonatal deaths. Previous studies have shown variability in mortality rates by healthcare setting. Also, different predictors influence the risk of neonatal mortality in different populations. Understanding the predictors of preterm neonatal mortality in the low-resource setting where we conducted our study could guide the development of interventions to improve outcomes for preterm neonates. We thus aimed to determine the incidence and predictors of mortality among preterm neonates born at Mbarara Regional Referral Hospital (MRRH) in South Western Uganda.

### Methods

We prospectively enrolled 538 live preterm neonates born at MRRH from October 2019 to September 2020. The neonates were followed up until death or 28 days, whichever occurred first. We used Kaplan Meier survival analysis to describe preterm neonatal mortality and Cox proportional hazards regression to assess predictors of preterm neonatal mortality over a maximum of 28 days of follow up.

### Results

The cumulative incidence of preterm neonatal mortality was 19.8% (95% C.I: 16.7–23.5) at 28 days from birth. Birth asphyxia (adjusted hazard ratio [aHR], 14.80; 95% CI: 5.21 to 42.02), not receiving kangaroo mother care (aHR, 9.50; 95% CI: 5.37 to 16.78), delayed initiation of breastfeeding (aHR, 9.49; 95% CI: 2.84 to 31.68), late antenatal care (ANC) booking (aHR, 1.81 to 2.52; 95% CI: 1.11 to 7.11) and no ANC attendance (aHR, 3.56; 95% CI:

**Funding:** The research reported in this manuscript was supported by The First Mile community Health Program at Mbarara University of Science and Technology. The content is solely the responsibility of the authors. The funders had no role in the study design, data collection, and analysis, decision to publish, or preparation of the manuscript.

**Competing interests:** The authors declare no competing interests.

1.51 to 8.43), vaginal breech delivery (aHR, 3.04; 95% CI: 1.37 to 5.18), very preterm births (aHR, 3.17; 95% CI: 1.24 to 8.13), respiratory distress syndrome (RDS) (aHR, 2.50; 95% CI: 1.11 to 5.64) and hypothermia at the time of admission to the neonatal unit (aHR, 1.98; 95% CI: 1.18 to 3.33) increased the risk of preterm neonatal mortality. Attending more than 4 ANC visits (aHR, 0.35; 95% CI: 0.12 to 0.96) reduced the risk of preterm neonatal mortality.

## Conclusions

We observed a high cumulative incidence of mortality among preterm neonates born at a low-resource regional referral hospital in Uganda. The predictors of mortality among preterm neonates were largely modifiable factors occurring in the prenatal, natal and postnatal period (lack of ANC attendance, late ANC booking, vaginal breech delivery, birth asphyxia, respiratory distress syndrome, and hypothermia at the time of admission to the neonatal unit, not receiving kangaroo mother care and delayed initiation of breastfeeding). These findings suggest that investment in and enhancement of ANC attendance, intrapartum care, and the feasible essential newborn care interventions by providing the warm chain through kangaroo mother care, encouraging early initiation of breastfeeding, timely resuscitation for neonates when indicated and therapies reducing the incidence and severity of RDS could improve outcomes among preterm neonates in this setting.

## Introduction

Preterm birth refers to a birth occurring at gestational age less than 37 completed weeks [1, 2]. It is an important cause of mortality and morbidity in new-borns [3]. Globally, in 2014 alone, about 15 million live preterm babies were born and the majority (81%) of these occurred in Asia and sub-Saharan Africa [4]. Preterm birth-related complications account for about 35% of all neonatal deaths [4]. In Uganda, the risk of preterm birth is estimated at 10 to 15%, and preterm birth is the leading cause of neonatal deaths, accounting for 28–31% of all neonatal deaths [5, 6].

Owing to the immature and inadequate physiological compensatory responses to the extra-uterine environment, premature babies are at an increased risk of complications such as respiratory distress syndrome, hypothermia, perinatal asphyxia, neonatal sepsis, intraventricular haemorrhage, neonatal jaundice, necrotizing enterocolitis and feeding difficulties [7, 8]. Unsurprisingly, the mortality from preterm births in resource-limited settings like Uganda is high since these complications are difficult to treat [6]. Interventions to prevent the occurrence of preterm birth like antenatal care attendance and administration of medicines that block the labour process are thus needed and studies focussing on this should be encouraged. Where preterm birth has occurred, interventions to prevent the development of complications that often lead to death are also urgently needed. In the long term, survivors of preterm birth are also more likely to experience motor and sensory impairment, delay in cognitive development and behavioural problems than babies born at term [6]. Long term monitoring of preterm babies and studies to guide such efforts are also important.

Deaths in the neonatal period are linked to maternal-fetal conditions and the care given during antepartum, intrapartum, and postpartum periods [9]. Maternal and obstetric factors implicated in preterm neonatal mortality include maternal age, increased body mass index

(BMI), diabetes mellitus, preeclampsia/eclampsia, smoking during pregnancy, preterm premature rupture of membranes (PPROM), antepartum haemorrhage (APH) and labour dystocia [10–14]. Neonatal factors contributing to mortality include low gestational age, low birth weight, 5-minute Apgar score less than 7 and the need for neonatal intensive care unit (NICU) admission [10–16]. Previous studies have shown variability in the mortality rates and outcomes in different health care settings and across countries; low resource settings like Uganda have higher mortality rates [17, 18].

A better understanding of the role of potentially modifiable risk factors like antenatal corticosteroid (ACS) administration, quality antenatal care (ANC), mode of delivery, tocolysis, respiratory support, kangaroo mother care (KMC), and postnatal surfactant may improve preterm neonatal outcomes, especially in low-resource settings [11, 15, 19]. This study aimed to determine the incidence and predictors of preterm neonatal mortality at Mbarara Regional Referral Hospital (MRRH), a low-resource hospital in South Western Uganda.

## Methods

### Study design and setting

We conducted a prospective cohort study of live preterm neonates born at Mbarara Regional Referral Hospital (MRRH), in South Western Uganda from October 2019 to September 2020. MRRH is a public tertiary hospital with a maternity ward that conducts approximately 10,000 deliveries per year, with a caesarean delivery rate of 40%, maternal mortality ratio of 375 per 100,000 live births [20] and perinatal mortality rate of 33 per 1000 live birth according to 2019 hospital records [21]. According to hospital records, routine antenatal care attendance averages at 1350 women per month. The maternity ward is managed by a team of about 14 obstetricians/gynaecologists, 38 obstetrics/gynaecology residents, intern doctors, and 38 midwives. There are approximately 50 preterm neonates delivered per month and these are usually transferred to the Paediatric neonatal unit within 10 minutes of delivery, except when there is need for resuscitation before transfer in which case a delay of up to 30 minutes may happen.

The Paediatric ward is managed by a team of about 8 paediatricians, 13 paediatric residents, intern doctors and 10 nurses. The Paediatric department admits about 5,000 children every year, approximately 2,000 (40%) of whom are neonates. Of the neonates admitted to the neonatal unit, approximately two-thirds are born at MRRH and a third are either referred in from other health facilities or directly from the surrounding communities. Premature babies make up 45% of all new-born admissions. The current neonatal unit accommodates 40 neonates. There is no separate KMC ward. Clinical care is provided by two paediatricians, a senior resident and an intern doctor alongside the 3 nurses. The neonatal unit has 2 neonatal phototherapy machines which serve 3–4 neonates at a time, and one full time functional radiant warmer. There is a supply of medical oxygen with oxygen cylinders, and 2 backup oxygen concentrators, 3 nasal continuous positive airway pressure (nCPAP) machines. There is an additional warmer in the emergency admission room.

### Participant enrolment

All live preterm neonates born at a gestational age from 28 weeks, 0 days (28W0D) to 36 weeks, 6 days (36W0D) at the MRRH maternity ward during the study period were eligible for the study. The gestational age (GA) was estimated using either the first day of the last normal menstrual period (LNMP), known to be a more reliable measure of GA in a low-resource setting [22] or first trimester dating obstetric ultrasound if available. Mothers with neither a known LNMP nor a first trimester ultrasound scan were excluded.

## Sample size determination and sampling techniques

According to the 2018 audit report from the Department of Paediatrics at MRRH (unpublished data), a total of 469 preterm neonates were admitted to the neonatal unit from a total of 3,636 new born babies. Seventy-five (75) of the 469 admitted preterm neonates died, yielding a cumulative mortality incidence of 16%. We used these data to estimate our expected total preterm births per month (≈40 neonates). We explored the precision that different sample sizes would yield around an estimated cumulative mortality of ≈17% for the 12 months planned study duration. The simulation result of 460 neonates would yield reasonable precision (about 2% length of the 95% CI) (Table 1). We therefore consecutively enrolled all live preterm neonates born during the study period while aiming to reach a minimum of 460 neonates.

## Data collection and follow-up

Mothers to the eligible preterm neonates were counselled before enrollment, after delivery. Those providing written informed consent were subsequently enrolled alongside their neonates. We developed and pretested the questionnaire to ascertain its adequacy to collect the study variables. The interviewer administered the questionnaire to obtain baseline data from mothers and followed this up with a chart review to obtain additional information about the mothers.

The data collected were categorised into: a) maternal characteristics (age, level of education, marital status, occupation, address, referral status (women referred to MRRH from another health center before delivery), and HIV status); b) obstetric characteristics (parity, gestation age at booking ANC visit, number of ANC visits, ACS administration, duration from administration of the first dose of ACS to delivery, mode of delivery, prior preterm birth, prior early neonatal death and obstetric conditions (preterm labour, PPROM, multiple pregnancy, pre-eclampsia/eclampsia, placenta previa or abruption placenta, and chorioamnionitis); and c) neonatal characteristics (neonatal sex, birth weight, gestational age at birth, and Apgar score at 5 minutes).

Preterm neonates, subsequently admitted to the neonatal unit, were followed up to ascertain outcomes through observation and chart review from admission time to death or until 28 days. Data abstracted from the neonatal charts included information on neonatal morbidities, interventions and, for those who died, the suspected cause of death. Neonatal morbidities considered were: respiratory distress syndrome (RDS), birth asphyxia, hypothermia, hypoglycaemia, neonatal sepsis, and necrotizing enterocolitis. Neonatal interventions assessed were: kangaroo mother care (KMC), supplemental oxygen therapy, nasal continuous positive airway pressure (nCPAP), prophylactic antibiotics, phototherapy, radiant heat warmed, nasogastric tube (NGT) feeding and breastfeeding initiation within the first hour of life. The clinical causes of neonatal death considered were: RDS, birth asphyxia, hypothermia, hypoglycaemia, neonatal sepsis, necrotizing enterocolitis, and nosocomial pneumonia.

For any neonates discharged alive before 28 days, post-discharge follow-up was done via a phone call on day 3, 7, 14 and 28. A fraction of the neonates discharged from the neonatal unit

**Table 1. Simulated samples sizes around the departmental cumulative neonatal mortality incidence.**

| N | Mortality estimate (95% CI) |
| --- | --- |
| 460 | 17% (15.0%-19.0%) |
| 480 | 17% (15.5%-18.5%) |
| 520 | 17% (15.7%-18.2%) |
| 540 | 17% (16.0%-18.0%) |

before 28 days were seen during scheduled post-discharge clinical visits at the neonatal clinic which operates once every week.

Data were collected into a tablet in some cases and a laptop in other cases and managed using a secure online backed up Research Electronic Data Capture (REDCap™) database (version 8.2) hosted at the Department of Obstetrics and Gynaecology at MUST [23].

## Study variables

The primary outcome was the time to death of a preterm neonate (in days) obtained via patient chart abstraction or phone call for deaths that occurred at home post-discharge. Primary predictors included maternal characteristics (age, level of education, marital status, occupation, address, referral status, and HIV status); obstetric characteristics (parity, gestation age at booking ANC visit, number of ANC visits, ACS administration, duration from administration of the first dose of ACS to delivery, mode of delivery, prior preterm birth, prior early neonatal death and obstetric conditions (preterm labour, PPROM, multiple pregnancy, pre-eclampsia/eclampsia, placenta previa or abruption placenta, and chorioamnionitis); and neonatal characteristics (neonatal sex, birth weight, gestational age at birth, and Apgar score at 5 minutes).

**Variable definitions.** A preterm infant was defined as a neonate born at a gestational age from 28 weeks, 0 days (28W0D) to 36 weeks, 6 days (36W6D) and further sub-classified according to gestational age as: very preterm (28weeks, 0days to 31weeks, 6days); moderate preterm (32weeks, 0days to 33weeks, 6days); and late preterm (34weeks, 0days to 36weeks, 6days). Birth weight was categorized into normal birth weight (>2.5 kg), low birth weight (1.5 to 2.5 kg), and very low birth weight (<1.5 kg). Antenatal corticosteroid use was defined as having received any dose of corticosteroids before delivery (within 7 days period of delivery). ANC booking—defined as the first time of ANC attendance, was categorized as: first trimester booking (<13 weeks of gestation), second trimester booking (between 13 to 26 weeks of gestation) and third trimester booking (≥ 27 weeks of gestation). We considered late ANC booking as ≥13 weeks of gestation [24].

Neonatal morbidities were defined as 1) RDS—presence of any of: fast breathing, grunting, subcostal and intercostal recession, cyanosis and reduced air entry in bilateral lung fields starting in the first four hours of life [25]; 2) birth asphyxia—5-minute Apgar score of ≤ 5 [26]; 3) hypothermia—axillary temperature of less than 36.0°C at the time of admission to the neonatal unit; 4) hypoglycaemia—glucose below 40mg/dL (<2.2mmol/l) in a capillary blood sample from a heel prick for blood glucose testing using an electronic glucometer [27]; 5) hyperbilirubinemia—yellowing of eyes and/or body requiring phototherapy or serum bilirubin levels above 15mg/dL as per WHO criteria [27]; 6) sepsis—the presence of clinical symptoms or signs suggestive of sepsis according to the WHO's Integrated Management of Childhood Illnesses (IMCI) algorithm with a blood culture at any point during the study period [27]; 7) nosocomial pneumonia—diagnosed basing on blood culture and localizing clinical features to the chest including grunting, fast breathing, chest wall in-drawing, oxygen saturation of <90% on room air, with crepitations on auscultation [27]; 8) necrotizing enterocolitis—diagnosed basing on clinical features of abdominal distension and tenderness, intolerance to feeding, bilious vomitus or fluid up the nasogastric tube, bloody stools and abdominal x-ray findings of air bubbles in the intestinal walls [27].

The cause of death was defined as the clinical condition directly and immediately leading to the death of the neonate as documented in the neonate's medical chart/records. The cause of death for those who died at home was classified as unknown.

## Statistical analysis

Data were exported from the REDCap ™ database to Stata (version 15) for cleaning and statistical analysis. Descriptive statistics were obtained for predictor and outcome variables and are presented in tables, charts and graphs. Kaplan Meier (KM) curves were used to estimate survival and describe the pattern of mortality among preterm neonates. Cox proportional hazards regression models were used to describe and assess the predictors of neonatal mortality among the preterm neonates in univariable and multivariable analyses. Preterm neonates that were lost to follow up were censored. However, the total time they contributed to the study was incorporated during analysis of results.

## Ethical consideration

Ethical approval was sought and obtained from the MUST Research Ethics Committee (MUREC-19/07-19) and the Uganda National Council for Science and Technology (UNCST-HS469ES). Written informed consent to participate in the study was obtained from the mothers before enrolment. All study methods were performed per the Declaration of Helsinki guidelines and regulations [28].

## Results

During the study period from October 2019 to September 2020, a total of 564 live preterm neonates were born at the maternity ward of MRRH. Of these, mothers to 26 neonates declined to consent, and 538 (95.4% response rate) were enrolled. Of the 538 preterm neonates, 106 died, 421 survived and 11 (2.0%) were lost to follow up by 28 days (Fig 1).

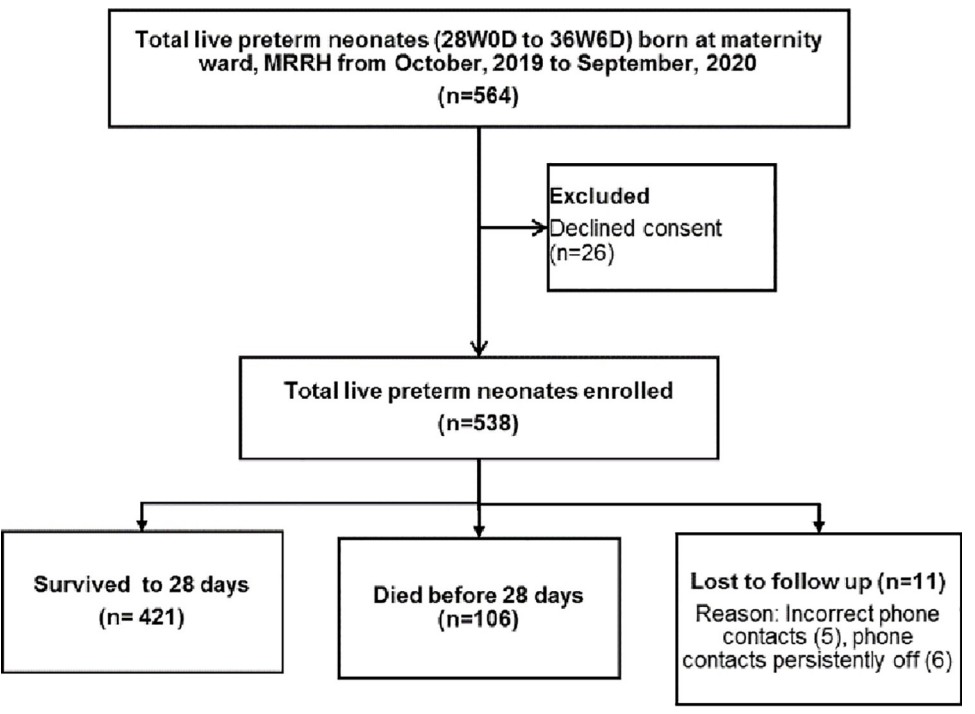

**Fig 1. Study flow diagram.**

## Maternal and obstetric characteristics

The mean maternal age was 26.4 (SD±5.9) years. Over half of the mothers (54.8%) lived in Mbarara district. The majority were married (97.0%), unemployed (69.5%), and had attained at least primary education (86.8%) (Table 2). Just under half of the sample were multiparous (49.1%). Of 505 (93.9%) mothers who attended ANC, 343 (63.8%) booked late and 290 (53.9%) attended 3–4 ANC visits. Of the 219 (40.7%) who received antenatal corticosteroids, 111 (50.7%) delivered within less than 24 hours after receiving the first dose. A majority (55.2%) of the neonates were delivered by spontaneous vaginal delivery. The majority of the preterm births followed spontaneous labour (66.5%), but significant proportions also occurred after some complications including PPROM (37.7%), and preeclampsia (11.9%) (Table 2).

## Neonatal characteristics of preterm neonates

Among 538 preterm neonates, 283 (52.6%) were males and269 (50%) were born at 34 weeks 0 days to 36 weeks 6 days. Three hundred one (56.0%) had low birth weight (LBW), while107 (19.9%) had very LBW. The majority were admitted to the neonatal unit (74.9%). Two hundred sixteen (37.5%) were diagnosed with respiratory distress syndrome (RDS), 148 (27.5%) with hypothermia at the time of admission, and 29 (7.2%) with birth asphyxia. Two hundred eighty (69.5%) received kangaroo mother care (KMC), 96 (23.8%) received continuous positive airway pressure (CPAP) (Table 3).

## Mortality incidence of preterm neonates

The cumulative incidence of preterm neonatal mortality was 9.1% at 24 hours, 13.2% at 72 hours, 17.6% at 7 days, 19.3% at 14 days, and 19.8% at 28 days. The cumulative preterm neonatal mortality incidence at 28 days in this study was therefore 19.8% (95% confidence interval: 16.7–23.5) (Fig 2).

## Neonatal morbidities contributing to death of preterm neonates

The morbidities contributing to death of preterm neonates included; respiratory distress syndrome (RDS) (71/106 [67.0%]), birth asphyxia (n = 17 [16.0%], hypothermia (n = 16 [15.1%]), neonatal sepsis (n = 11 [10.4%]), jaundice (n = 7 [6.6%]), necrotizing enterocolitis (NEC) (n = 4 [3.8%]), nosocomial pneumonia (n = 3 [2.8%]), and congenital anomalies (n = 3 [2.8%]). The contributing morbidities were classified as unknown for neonates that died at home after discharge (n = 6 [5.7%])—2 neonates were reported to have had a febrile illness prior to their death, while 4 neonates were reported to have been well but found dead in the bed (Fig 3).

## Predictors of mortality among preterm neonates

Birth asphyxia (aHR, 14.80; 95% CI: 5.21–42.02), not receiving KMC (aHR, 9.50; 95% CI: 5.37–16.78), late initiation of breastfeeding (>1 hour after birth) (aHR, 9.49; 95% CI: 2.84–31.68), not attending ANC (aHR, 3.56; 95% CI: 1.51–8.43), late ANC booking (booking in the second trimester (adjusted hazard ratio [aHR], 1.81; 95% CI: 1.03–3.64) and third trimester aHR, 2.52; 95% CI: 1.11–7.11)), vaginal breech delivery (aHR, 3.04; 95% CI: 1.37–5.18), very preterm births (aHR, 3.17; 95% CI: 1.24–8.13), RDS (aHR, 2.50; 95% CI: 1.11–5.64) and hypothermia at admission (aHR, 1.98; 95% CI: 1.18–3.33) increased the risk of preterm neonatal mortality. However, attending more than 4 ANC visits (aHR, 0.35; 95% CI: 0.12–0.96) reduced the risk of mortality (Table 4).

**Table 2. Baseline socio-demographic and obstetric characteristics of mothers of preterm neonates born at Mbarara Regional Referral Hospital from October 2019 to September 2020 (N = 538).**

| Characteristics | Frequency (n) | Percentage (%) |
|---|---|---|
| **Maternal residence** | | |
| Mbarara | 295 | 54.8 |
| Isingiro | 123 | 22.9 |
| Others | 120 | 22.3 |
| **Maternal age (years)** | | |
| <20 | 68 | 12.6 |
| 20–34 | 413 | 76.8 |
| >34 | 57 | 10.6 |
| **Married** | 522 | 97.0 |
| **Level of education** | | |
| Uneducated | 71 | 13.2 |
| Primary | 202 | 37.6 |
| Secondary | 180 | 33.5 |
| Post-secondary | 85 | 15.8 |
| **Employed** | 164 | 30.5 |
| **Mother referred in** | 255 | 47.4 |
| **HIV serostatus** | | |
| HIV positive | 86 | 16.0 |
| HIV negative | 452 | 84.0 |
| **Parity (number of births)** | | |
| I | 157 | 29.2 |
| II-IV | 264 | 49.1 |
| ≥V | 117 | 21.7 |
| **Booking (first) ANC visit** | | |
| 1st trimester | 162 | 30.1 |
| 2nd trimester | 307 | 57.1 |
| 3rd trimester | 36 | 6.7 |
| **ANC attendance** | | |
| Did not attend ANC | 33 | 6.1 |
| 1–2 times | 110 | 20.5 |
| 3–4 times | 290 | 53.9 |
| >4 times | 105 | 19.5 |
| **Mode of delivery** | | |
| Spontaneous vaginal delivery | 297 | 55.2 |
| Vaginal breech delivery | 39 | 7.3 |
| Emergency caesarean section | 182 | 33.8 |
| Elective caesarean section | 20 | 3.7 |
| **Antenatal corticosteroid (ACS) use** | 219 | 40.7 |
| **Corticosteroid use to delivery interval** | N = 219 | |
| < 24 hours | 111 | 50.7 |
| 24 to <48 hours | 79 | 36.1 |
| ≥48 hours | 29 | 13.2 |
| **Tocolysis** | 18 | 3.6 |
| **Previous preterm birth** | 14 | 2.6 |
| **Previous history of a still births** | 17 | 3.2 |
| **Previous history of ENND** | 22 | 4.1 |

(*Continued*)

**Table 2.** (Continued)

| Characteristics | Frequency (n) | Percentage (%) |
|---|---|---|
| **Obstetric conditions** | | |
| PPROM | 203 | 37.7 |
| Spontaneous preterm labor | 358 | 66.5 |
| Preeclampsia | 64 | 11.9 |
| APH (placenta previa or abruption) | 33 | 6.1 |
| Mal-presentation | 36 | 6.7 |
| Chorioamnionitis | 9 | 1.7 |
| Other obstetric conditions | 15 | 2.8 |

ANC—antenatal care, ACS—antenatal corticosteroids, ENND—early neonatal death, PPROM—preterm premature rupture of membranes, APH—antepartum haemorrhage.

**Table 3. Characteristics of preterm neonates born at Mbarara Regional Referral Hospital from October 2019 to September 2020 (N = 538).**

| Characteristics | Frequency (n) | Percentage (%) |
|---|---|---|
| **Sex of the neonate** | | |
| Male | 283 | 52.6 |
| Female | 255 | 47.4 |
| **Gestational age** | | |
| 28W0D-31W6D (very preterm) | 165 | 30.7 |
| 32W0D-33W6D (moderate preterm) | 104 | 19.3 |
| 34W0D-36W6D (late preterm) | 269 | 50.0 |
| **Birth weight (kg)** | | |
| <1.5 (ELBW & VLBW) | 107 | 19.9 |
| 1.5 - <2.5 (LBW) | 301 | 56.0 |
| ≥2.5 (Normal) | 130 | 24.2 |
| **Apgar score at 5 minutes** | | |
| < 7 | 56 | 10.4 |
| ≥7 | 481 | 89.6 |
| **Neonate required admission to neonatal unit** | 403 | 74.9 |
| **Neonatal morbidities** | | |
| Birth asphyxia | 25 | 4.6 |
| Respiratory Distress Syndrome (RDS) | 223 | 41.4 |
| Hypothermia at the time of admission | 148 | 27.5 |
| Hypoglycemia at the time of admission | 27 | 5.0 |
| Neonatal sepsis | 83 | 15.4 |
| Neonatal jaundice | 135 | 25.1 |
| Necrotizing enterocolitis (NEC) | 7 | 1.3 |
| Nosocomial pneumonia | 9 | 1.7 |
| **Neonatal interventions** | | |
| Kangaroo mother care (KMC) | 280 | 69.5 |
| Required and received continuous positive airway pressure (CPAP) | 96 | 23.8 |
| Required and received supplemental oxygen therapy | 295 | 73.2 |
| Required and received phototherapy | 141 | 35.7 |
| Warmed with radiant warmer | 380 | 94.3 |
| Nasogastric tube feeding | 162 | 40.3 |
| Breastfeeding initiated within the first hour of life | 129 | 31.9 |

ELBW—Extreme low birth weight, VLBW—very low birth weight, LBW—Low birth weight.

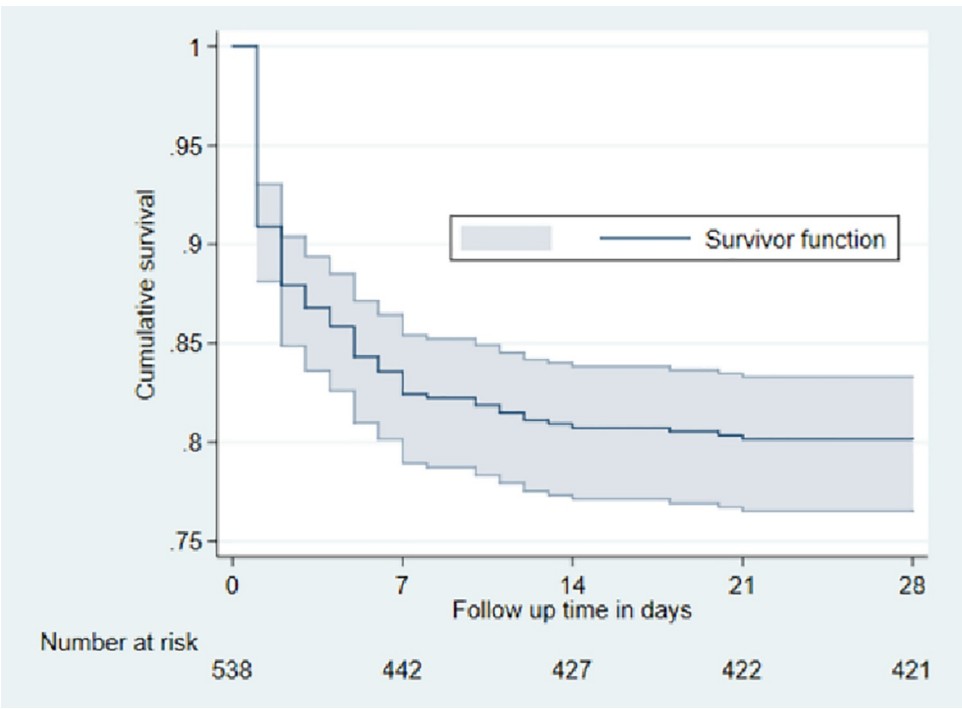

**Fig 2. Cumulative survival curve of preterm neonates born at MRRH from October, 2019 to September, 2020 (n = 538).**

## Discussion

This study aimed to determine the incidence and predictors of preterm neonatal mortality in a tertiary low-resource hospital in South Western Uganda. We observed a high overall incidence of mortality (19.8%). The study highlights the major predictors of preterm neonatal mortality

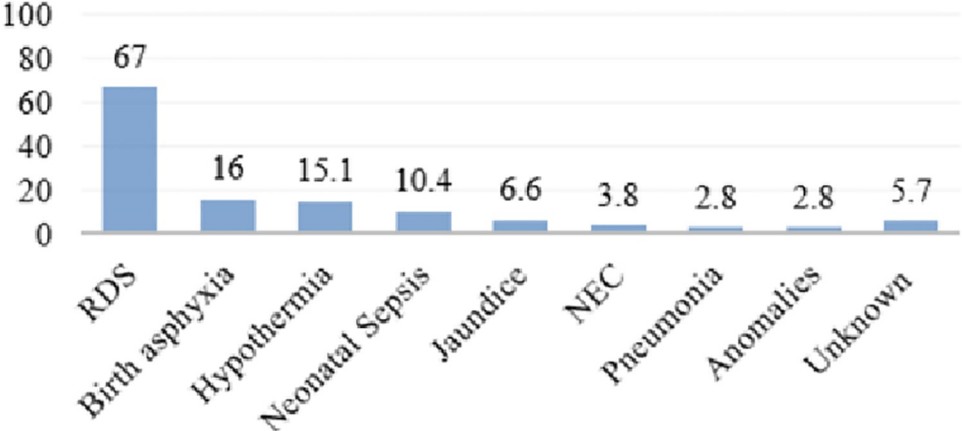

**Fig 3. Bar chart showing the clinical causes of death among preterm neonates born at MRRH from October, 2019 to September, 2020 (n = 106).** RDS—respiratory distress syndrome; NEC—necrotizing enterocolitis.

**Table 4. Predictors of mortality among preterm neonates born at Mbarara Regional Referral Hospital from October, 2019 to September, 2020.**

| Predictor variables | Categories | CHR (95% CI) | p-value | AHR (95% CI) | p-value |
|---|---|---|---|---|---|
| **Booking ANC visit** | 1st trimester | Ref. | | Ref. | |
| | 2nd trimester | 2.30 (1.31–4.04) | 0.004* | 1.81 (1.03–3.64) | 0.044* |
| | 3rd trimester | 5.22 (2.55–10.68) | <0.001** | 2.52 (1.11–7.11) | 0.032* |
| **Number of ANC visits** | 3–4 visits | Ref. | | Ref. | |
| | No ANC attended | 2.68 (1.48–4.83) | 0.001* | 3.56 (1.51–8.43) | 0.004* |
| | 1–2 visits | 1.78 (1.15–2.75) | 0.010* | 0.76 (0.41–1.42) | 0.390 |
| | >4 times | 0.35 (0.16–0.78) | 0.010* | 0.35 (0.12–0.96) | 0.042* |
| **Mode of delivery** | Caesarean delivery | Ref. | | Ref. | |
| | Spontaneous vertex delivery | 0.90 (0.58–1.40) | 0.642 | 1.06 (0.62–1.79) | 0.833 |
| | Vaginal breech delivery | 5.66 (3.40–9.42) | <0.001** | 3.04 (1.37–5.18) | 0.004* |
| **ACS use** | Yes | Ref. | | Ref. | |
| | No | 2.87 (1.79–4.59) | <0.001* | 1.01 (0.56–1.82) | 0.966 |
| **Gestational Age** | 34W0D-36W6D | Ref. | | Ref. | |
| | 32W0D-33W6D | 1.43 (0.71–2.88) | 0.323 | 1.38 (0.51–3.74) | 0.525 |
| | 28W0D-31W6D | 6.20 (3.84–10.00) | < 0.001** | 3.17 (1.24–8.13) | 0.016* |
| **Birth weight (kg)** | ≥2.5 | Ref. | | Ref. | |
| | 1.5-<2.5 | 2.26 (1.06–4.82) | 0.035* | 2.30 (0.54–9.79) | 0.258 |
| | <1.5 | 10.0 (4.73–21.14) | < 0.001** | 3.20 (0.66–15.50) | 0.148 |
| **RDS** | No | Ref. | | Ref. | |
| | Yes | 6.59 (4.09–10.63) | <0.001** | 2.50 (1.11–5.64) | 0.027* |
| **Hypothermia at admission** | No | Ref. | | | |
| | Yes | 4.38 (2.97–6.46) | <0.001** | 1.98 (1.18–3.33) | 0.010* |
| **Birth asphyxia** | No | Ref. | | Ref. | |
| | Yes | 8.03 (4.88–13.22) | <0.001** | 14.80 (5.21–42.02) | < 0.001** |
| **Hypoglycemia** | No | Ref. | | Ref. | |
| | Yes | 2.14 (1.12–4.11) | 0.022* | 1.92 (0.91–4.03) | 0.087 |
| **NEC** | No | Ref. | | Ref. | |
| | Yes | 3.08 (1.13–8.37) | 0.027* | 0.43 (0.14–1.28) | 0.128 |
| **KMC** | Yes | Ref. | | Ref. | |
| | No | 7.82 (4.95–12.35) | <0.001** | 9.50(5.37–16.78) | <0.001** |
| **Breastfeeding initiation at birth** | Early (≤ 1 hour) | Ref. | | Ref. | |
| | Late (>1 hour) | 5.51 (3.28–9.26) | <0.001** | 9.49(2.84–31.68) | <0.001** |
| **nCPAP** | No | Ref. | | Ref. | |
| | Yes | 6.22 (4.08–9.49) | <0.001** | 0.80(0.44–1.45) | 0.466 |
| **Supplemental oxygen** | No | Ref. | | Ref. | |
| | Yes | 2.86 (1.53–5.38) | 0.001* | 1.22 (0.46–3.25) | 0.686 |
| **Neonatal Sex** | Male | Ref. | | Ref. | |
| | Female | 0.73 (0.50–1.08) | 0.112 | 0.63 (0.39–1.01) | 0.053 |

CHR—crude hazard ratio, AHR—adjusted hazard ratio, *p<0.05, **p <0.001,ANC—antenatal care, ACS—antenatal corticosteroid, RDS—respiratory distress syndrome, NEC—necrotizing enterocolitis, KMC—kangaroo mother care, nCPAP—nasal continuous positive airway pressure.

such as birth asphyxia, not receiving kangaroo mother care (KMC), late initiation of breast-feeding, late ANC booking and lack of ANC attendance, vaginal breech delivery, very preterm birth, respiratory distress syndrome and hypothermia at the time of admission to the neonatal unit as increasing the risk of preterm neonatal mortality. Attending more than four ANC visits decreased the risk of preterm neonatal mortality.

The observed incidence of neonatal mortality is comparable to findings from other studies done in similar resource limited settings. For instance, a study done in Mulago National Referral Hospital in Central Uganda reported 22.1% [29], while a multicentre prospective observational study at 5 hospitals in Ethiopia reported 22.7% [30]. Also findings from the University of Nigeria Teaching Hospital in Nigeria showed a mortality incidence of 24.0% [31]. This is likely because of the similar vulnerable study population of preterm neonates, but also the comparable study settings which were referral teaching hospitals. Our result is higher than what was observed at Nairobi hospital in Kenya (11.9%) [32], and Shahid Akbar-Abadi university hospital in Iran (9.1%) [11]. However, our finding is lower than what has been observed at other regional referral hospitals in Uganda: 35.2% in Kiwoko hospital in Central Uganda [33]; and 31.6% in Fort Portal Regional Referral Hospital in Western Uganda [34]. Our result is also lower than what was observed at the NICU in North West Ethiopia (28.8%) [35],and the Fatemieh hospital in Iran (27.4%) [36].

The disparity in mortality is likely due to inequalities in neonatal care for preterm neonates. Settings with specialized and well-equipped neonatal care facilities could provide better care to the preterm neonates compared to the resource limited settings with suboptimal care provided in the inadequately equipped NICUs as noted in sub-Saharan Africa [37]. The differences could also be explained by the source of data. At Nairobi hospital in Kenya and Shahid Akbar-abadi university hospital in Iran (where lower mortality was reported) data was obtained for neonates born at the respective hospitals. At Kiwoko hospital in Uganda, Fort Portal Regional Referral Hospital in Uganda, University of Gondar Comprehensive Specialized Hospital in Ethiopia and Fatemieh hospital in Iran, data were obtained from both delivery and neonatal intensive care units. The studies included neonates referred in from other centres or born from home, and also enrolled extremely preterm neonates (<28 weeks of gestation). Our study prospectively followed up preterm neonates from the delivery room to neonatal unit and / or home, and only enrolled preterm neonates with gestational age at least 28 weeks to <37 weeks born from the maternity unit of Mbarara Regional Referral Hospital, which may explain why our observed mortality was lower than that observed at the other Ugandan hospitals.

The main contributors to neonatal death established in this study were respiratory distress syndrome (RDS), birth asphyxia, hypothermia and neonatal sepsis. This is similar to what is reported in other studies [29, 30, 38]. Respiratory distress syndrome, birth asphyxia and hypothermia are common complications of preterm neonates. Respiratory distress syndrome is associated with surfactant deficiency which is common with decreasing gestational age, mainly in neonates born before 34 weeks of gestation [39]. Hypothermia is mainly due to the large body surface area to weight and the relative lack of subcutaneous fat which makes the preterm neonates prone to heat loss [40]. Notably, neonatal sepsis contributed substantially to mortality, despite preterm neonates, especially those that required admission, receiving prophylactic antibiotics.

The predictors of mortality among preterm neonates born at MRRH were largely modifiable and ranged from prenatal, natal and postnatal predictors (lack of ANC attendance or late ANC booking, vaginal breech delivery, birth asphyxia, respiratory distress syndrome and hypothermia at the time of admission to the neonatal unit, not receiving kangaroo mother care and delayed initiation of breastfeeding). Similar findings have been reported from other studies of various settings [34, 35, 41–47]. The predictors found in our study suggest that relatively simple interventions might improve the outcomes of the preterm neonates born at MRRH. Investment in and enhancement of the feasible essential new born (ECN) interventions which include providing the warm chain through kangaroo mother care, encouraging early initiation of breastfeeding and timely resuscitation for neonates when indicated could improve outcomes among preterm neonates [48]. Also strategies to reinforce quality ANC

attendance, vigilant intrapartum monitoring and appropriate choice of the delivery mode specifically avoiding vaginal delivery for breech preterm neonates, enhanced care for especially the very preterm neonates at greatest risk for preterm-associated complications and mortality mostly within 7 days of life, and scaling up the implementation of therapies that reduce incidence and severity of respiratory distress syndrome (antenatal corticosteroid administration, use of CPAP and postnatal surfactant) is warranted.

The strength of this study was that it was a prospective study conducted at a single tertiary centre with a large number of preterm neonates staying in the same centre from birth to discharge with a uniform policy on labour room practices, management protocols and discharge criteria. We followed up neonates from the delivery room to the neonatal unit and home until the end of the neonatal period. The discharged preterm neonates were followed up via phone to establish their outcome on days 3, 7, 14 and 28. However, since we only enrolled and followed up neonates born at MRRH, the mortality incidence may not reflect the overall preterm neonatal mortality in the community. Some neonates were lost to follow-up due to persistently off or incorrect telephone contacts provided by study participants. To address this limitation, we had obtained more than one phone contact and we also followed up and tracked some of the lost neonates at the neonatal out-patient clinic that runs every Thursday at the paediatrics ward. This helped to reduce the attrition rates but still, there were a few neonates for whom outcomes could not be established. Since the first trimester obstetric ultrasonography is not routinely accessible to most pregnant women, we used LNMP to determine the gestational age. Also, nosocomial pneumonia was reported separate from neonatal sepsis to highlight the observed contribution of chest infections to the overall neonatal sepsis at our institution. No additional intervention was available for neonates with RDS who failed on CPAP, this could have increased the contribution of RDS to the overall mortality. Confidence intervals for hazard ratios of birth asphyxia and late initiation of breastfeeding were wide, and should therefore be interpreted in context of small numbers of participants with birth asphyxia and those with early initiation of breastfeeding. Finally, although we suggest various clinical causes of death, postmortem examinations were not done. We based on the clinical cause of death as documented by the clinical care team, which can at times be inaccurate.

## Conclusions

The preterm neonatal mortality observed in this study was high, especially among the very preterm neonates. The majority of the deaths occurred in the early neonatal period, and the leading clinical causes of death were respiratory distress syndrome, birth asphyxia and hypothermia. The predictors for mortality are largely modifiable including obstetric predictors like lack of ANC attendance, late ANC booking, attending more than 4 ANC visits and vaginal breech delivery, and neonatal predictors like very preterm births, neonatal morbidities suffered including respiratory distress syndrome, hypothermia and birth asphyxia, not receiving KMC and late initiation of breastfeeding.

Our findings highlight the importance of integrated maternal-neonatal care for preterm neonates. Interventions designed around the modifiable predictors such as reinforcing quality ANC attendance, vigilant intrapartum monitoring and appropriate choice of the delivery mode specifically avoiding vaginal delivery for breech preterm neonates could improve outcomes. There is also need for enhanced care for, especially the very preterm neonates at greatest risk for preterm-associated complications and mortality mostly within the early neonatal period. Scaling up the implementation of therapies that reduce incidence and severity of respiratory distress syndrome like antenatal corticosteroid administration, use of CPAP and postnatal surfactant, improving thermal care through kangaroo mother care and encouraging early

initiation of breastfeeding could improve outcomes among preterm neonates. We recommend future studies to evaluate the feasibility of implementing these interventions and what their impact would be on preterm neonatal outcomes in this setting.

## Supporting information

**S1 Fig.**
(TIF)

**S2 Fig.**
(TIF)

**S3 Fig.**
(TIF)

**S4 Fig.**
(TIF)

**S5 Fig.**
(TIF)

**S6 Fig.**
(TIF)

**S7 Fig.**
(TIF)

**S1 Dataset.**
(DTA)

## Acknowledgments

We acknowledge the research assistants Ms. Shanura Masika and Ms. Edivina Ahimbisibwe. We are grateful to the staff at the maternity ward and neonatal unit of Mbarara Regional Referral Hospital, Mbarara University of Science and Technology and the participants for enabling the performance of this study.

## Author Contributions

**Conceptualization:** Leevan Tibaijuka, Stephen M. Bawakanya, Joseph Ngonzi, Stephen B. Asiimwe, Godfrey R. Mugyenyi.

**Data curation:** Leevan Tibaijuka, Asiphas Owaraganise, Stephen B. Asiimwe.

**Formal analysis:** Leevan Tibaijuka, Asiphas Owaraganise, Stephen B. Asiimwe.

**Funding acquisition:** Leevan Tibaijuka.

**Investigation:** Leevan Tibaijuka, Stephen M. Bawakanya, Godfrey R. Mugyenyi.

**Methodology:** Leevan Tibaijuka, Stephen M. Bawakanya, Asiphas Owaraganise, Lydia Kyasimire, Elias Kumbakumba, Adeline A. Boatin, Musa Kayondo, Joseph Ngonzi, Godfrey R. Mugyenyi.

**Project administration:** Leevan Tibaijuka.

**Writing – original draft:** Leevan Tibaijuka, Stephen M. Bawakanya, Asiphas Owaraganise, Lydia Kyasimire, Elias Kumbakumba, Adeline A. Boatin, Musa Kayondo, Joseph Ngonzi, Stephen B. Asiimwe, Godfrey R. Mugyenyi.

**Writing – review & editing:** Leevan Tibaijuka, Stephen M. Bawakanya, Asiphas Owaraganise, Lydia Kyasimire, Elias Kumbakumba, Adeline A. Boatin, Musa Kayondo, Joseph Ngonzi, Stephen B. Asiimwe, Godfrey R. Mugyenyi.

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
