## [Decision Letter · Decision Letter 0]

3 Aug 2021

PONE-D-21-19132

Incidence and predictors of preterm neonatal mortality at Mbarara Regional Referral Hospital in South Western Uganda

PLOS ONE

Dear Dr. Tibaijuka

Thank you for submitting your manuscript to PLOS ONE. After careful consideration, we feel that it has merit but does not fully meet PLOS ONE’s publication criteria as it currently stands. Therefore, we invite you to submit a revised version of the manuscript that addresses the points raised during the review process.

Thank you for your interesting article. Please could you address the concerns from the 2 reviewers and re-submit.

We look forward to receiving your revised manuscript.

Kind regards,

Lloyd J. Tooke, MBChB, MMed(Paeds)

Academic Editor

PLOS ONE

Journal Requirements:

2. We note that you have referenced (ie. Bewick et al. [5]) which has currently not yet been accepted for publication. Please remove this from your References and amend this to state in the body of your manuscript: (ie “Bewick et al. [Unpublished]”) as detailed online in our guide for authors

Reviewers' comments:

Reviewer's Responses to Questions

**Comments to the Author**

1. Is the manuscript technically sound, and do the data support the conclusions?

Reviewer #1: Yes

Reviewer #2: Yes

2. Has the statistical analysis been performed appropriately and rigorously? 

Reviewer #1: Yes

Reviewer #2: I Don't Know

3. Have the authors made all data underlying the findings in their manuscript fully available?

Reviewer #1: Yes

Reviewer #2: Yes

4. Is the manuscript presented in an intelligible fashion and written in standard English?

Reviewer #1: Yes

Reviewer #2: Yes

5. Review Comments to the Author

Reviewer #1: PLOS One – Comments to the Author

1. This is an important study on incidence of preterm neonatal mortality and predictors of mortality in a tertiary hospital in South Western Uganda. It highlighted the major predictors such as low antenatal attendance, Hypothermia and RDS and lack of Kangaroo mother care. The predicators identified could help to design appropriate interventions to reduce the preterm related Mortality in the Tertiary Insusitution.

2. The Abstract is well summarised , I would suggest improve the language in paragraph 1

3. The statistics is Rigorous and appropriate for the Study design

4. METHODS SECTION

5. Did we exclude the extreme low birth weight , were all the babies at 28 weeks one kilo and above , is that true, what of the babies born to PET mothers , Did we exclude the extreme low birth weight, if so why, this may affect the mortality rates.

It is surprising all the babies were all above one kilo

6. Did we do Bilirubin levels done, or did we try to use a Kramer Chart – Please clearly state the method of measurement of the Bilirubin Levels

7. The babies that had Respiratory distress sydromme, was it all in the first week

8. What was their time frame ( Between first week ) or was this involving those after one week , If after one week how would you differentiate between other diseases like sepsis or apnea of Prematurity from Respiratory Distress syndrome, because RDS seems to account for many Deaths and yet diseases like sepsis , donot account for very few deaths, which is surprising

9. How would you diagnose Nosocomial pneumonia, were blood Cultures done , What do you mean , Could that be late one set sepsis

10. When was the Hypothermia Diagnosed at the Time of admission or during the course of admission

Discussion

11. Introduction could be improved , by highlighting the purpose of the study, instead of reporting all the important significant Results in the First Paragraph

12. The Respiratory Distress sydromme , assessment, we would have used the SAS score , it has been used in Kiwoko Hospital

13. When did the Hypothermia Occur, was it birth or during the admission, because hypothermia on admission to the NICu is associated with mortality. This could be added to the explanation for hypothermia

14. The difference in the mortality could be explained by the Different levels of care in neonatology ( refer to article Survive and thrive, Inpatient care for small and sick newbor barriers By Hannah Blencowe 2015 )

15. The different interventions available in Mbarara , how do they differ with other the hospitals

16. RDS – major cause of mortality , more studies from Mulago could explain that ( Yaser et al 2012)

17. More explanation for Hypothermia, what prevents Hypothermia, do we have a warm chain, what interventions are available for Hypothermia apart from kangaroo mother care

18. Need to explain, why those that needed CPAP, Died , do we have a criteria for those that have failed CPAP, if so what do we do , do we refer , we need to explain in the discussion

Reviewer #2: Introduction:

This section is well written, however to increase clarity, in this sentence “It is stated that Interventions to prevent the occurrence of preterm birth are thus needed and studies focusing on this should be encouraged”, it would be useful to mention a few, since there are some of spontaneous preterm births with no cause which may not be easy to prevent.

Method:

Study site;

The section is well described; my few comments and questions are as follows

Please add information on the ANC attendance pattern in the study site, also availability of KMC ward and size.

Participant enrollment:

•Was the questionnaire used a newly developed by the researchers or an adopted version of questionnaire which is validated?

Data collection and follow up

•What time was the baseline information collected, was it after delivery or after arrival to newborn unit, this did not come out clearly.

•Authors stated that they used the first day of the last normal menstrual period (LNMP), known to be a more reliable measure of GA in a low-resource setting, however this is prone to recall bias especially if the literacy level is low, this has been a recurring problem in our unit as well, thus this can be cited as a limitation to the study. The use of Modified Ballard score could have added value.

•Suspected cause of death was used, why not actual cause of death since only deaths which occurred in hospital were analysed.

•For any neonates discharged alive before 28 days, post-discharge follow-up was done via a phone call at day 3, 7, 14 and 28, What was the reasons for frequent phone calls, is this part of standard of care? If not was this included in the consent form signed by parents/participants; please clarify.

•For those who died after discharge, one would have expected the deaths which occurred at home to have had verbal autopsy done to establish cause of death, this could have been possible since the deaths were confirmed by mobile phone. Was there a reason this was omitted?

Study variables:

•It would be very useful if the author could clearly clarify the followings in the definition of variables.

-Hypothermia: was this at admission or at any point after admission

-Sepsis: was this combining EOS and LOS ie sepsis diagnosis at any point?

-Hypoglycemia: at admission or any time after, also a cut off point of 2.2 was used, which seems too low, is there an explanation? WHO refers to <2.5mmol/L. also note there is a difference between plasma and blood glucose thus it is always useful to state clearly.

-ACS: any dose at any time before delivery?

-How was RDS differentiated from congenital pneumonia? Especially for mothers with risk factors. -There is also typo instead of starting, it is written staring

-What was the cut off for booking for ANC late? 2nd trimester third, please clarify.

Analysis:

•Were the interaction between variables examined in the Cox proportional hazards regression models; some of the factors seems to be correlated and can affect the outcomes. Ie a baby with sepsis may also present with hypothermia, jaundice. Also including both Birth asphyxia and Apgar score in the same model can cancel the effects. I would advice the Authors to check for correlation and interaction in the covariates included in the final models.

Results:

•Arrangement and labeling of tables and figures is a bit confusing. Ie what appears as table 1 in the result section I presume it should be labeled as table 3, please check.

•For all the tables and figures please include a footnote spelling out all the abbreviations used.

•The KM figures are not clear, title shows mortality probabilities, but the Y –axis shows cumulative survival, please check and rectify.

•It seems like very few-used CPAP (Ie 216 were diagnosed with RDS but only 96 received CPAP) was it because of availability? It would be informative for policy if the median survival time was shown among those with RDS who had access to CPAP Vs those who had no access.

• Under neonatal characteristics the proportion of babies with Sepsis is not shown.

•Neonatal morbidities contributing to death of preterm neonates, I am not sure if there is a clear line to differentiate NEC, nosocomial pneumonia and sepsis, it will be informative to the readers if the authors could say how they managed to do so.

•It is also important to specify of sepsis was based on symptoms or cultures, since in the definition of variable it was either or

•Preterm are prone to IVH and anemia, which contributes to increased mortality, were any of these observed in this study?

•Figure 2, I would advise to use only one curve, ie cumulative survival curve.

•Figure 2; the chart area will be seen more clearly if the font size for the texts was reduced.

•As much as results presented graphically are more visual, having a lot of KP curves around predictors is tiring. For preterm it will be more informative if the median survival time by infant demographics eg sex, GA (late, moderate and early preterm) or weight (LBW/SGA vs LBW/AGA)

•Figure 10 Need for ventilatory support with CPAP, it is obvious that those who needed CPAP may have been versus sick and thus have increased mortality. For policy implication it would be more meaningful to show if those who needed and had access vs those who needed but did not have an access

•I suggest table 4 be omitted, since What is presented in table 4 is a duplicate of what is presented in figure 2,

Predictors of mortality among preterm neonates;

•For strong predictors it will be useful to focus more on those with high aHR and significant CI. Eg Birth asphyxia (aHR 11.9), not receiving KMC(aHR 9.14), late initiation of BF (aHR 8.56), based on these results suggests investment in essential newborn care could reduce mortality in these babies

•Some of the confidence intervals are very wide ie birth asphyxia (aHR, 11.90; 95% CI: 4.08-34.70). Can the authors comment on this and the implication it has on their results and recommendations.

Discussion:

•It would have been more informative if the author could discuss in relation to feasible interventions to overcome the problems they identified. Ie based on the strong predictors enhancement of ENC should be more discussed, I guess many investors would give the money to improve this over postnatal surfactant, which has less value if ENC is not adequate.

References: 19, 37,40 and 42 need to be appropriately cited, they should clearly indicate the source; if they are from the internet then URL and date retrieved should be included.

6. PLOS authors have the option to publish the peer review history of their article (what does this mean?). If published, this will include your full peer review and any attached files.

Reviewer #1: No

Reviewer #2: No

---

## [Author Response · Author response to Decision Letter 0]

23 Aug 2021

RESPONSE TO ACADEMIC EDITOR’S COMMENTS

Response: The manuscript has been reformatted to be in accordance with the PLOS ONE style

2. We note that you have referenced (ie. Bewick et al. [5]) which has currently not yet been accepted for publication. Please remove this from your References and amend this to state in the body of your manuscript: (ie “Bewick et al. [Unpublished]”) as detailed online in our guide for authors

Response: The reference list has been reviewed, we have ensured it is complete and correct.

RESPONSE TO REVIEWERS’ COMMENTS

3. This is an important study on incidence of preterm neonatal mortality and predictors of mortality in a tertiary hospital in South Western Uganda. It highlighted the major predictors such as low antenatal attendance, Hypothermia and RDS and lack of Kangaroo mother care. The predicators identified could help to design appropriate interventions to reduce the preterm related Mortality in the Tertiary Institution. 

Response: We appreciate the reviewer for the positive comments. 

4. The Abstract is well summarised, I would suggest improve the language in paragraph 1

Response: We have revised the language in the first paragraph to improve clarity. It now reads, “Preterm neonatal mortality contributes substantially to the high neonatal mortality globally. In Uganda, preterm neonatal mortality accounts for 31% of all neonatal deaths” (Page/Line: 2/20-21)

5. The statistics is Rigorous and appropriate for the Study design 

Response: Thanks for the comment.

METHODS SECTION

6. Did we exclude the extreme low birth weight , were all the babies at 28 weeks one kilo and above , is that true, what of the babies born to PET mothers , Did we exclude the extreme low birth weight, if so why, this may affect the mortality rates. 

It is surprising all the babies were all above one kilo 

Response: Thank you for raising this concern. We had very few extreme LBW babies (11) and analysed them in the category of VLBW babies (<1.5kg). (Table 3: Page 13)

7. Did we do Bilirubin levels done, or did we try to use a Kramer Chart – Please clearly state the method of measurement of the Bilirubin Levels

Response: We combined both clinical and laboratory techniques to assess serum bilirubin levels including yellow discoloration of eyes, skin and mucous membranes, and serum bilirubin levels above 15mg/dL. (Page/line: 6/183-184)

8. The babies that had Respiratory distress syndrome, was it all in the first week

Response: Yes. Respiratory distress syndrome (RDS) occurred in the first week. 

9. What was their time frame ( Between first week ) or was this involving those after one week , If after one week how would you differentiate between other diseases like sepsis or apnea of Prematurity from Respiratory Distress syndrome, because RDS seems to account for many Deaths and yet diseases like sepsis , do not account for very few deaths, which is surprising

Response: Same as in #8 above, the diagnosis of RDS was made within 4 hours of delivery, none was observed after one week.

10. How would you diagnose Nosocomial pneumonia, were blood Cultures done , What do you mean , Could that be late one set sepsis 

Response: Nosocomial pneumonia was diagnosed basing on blood culture and localizing clinical features to the chest/ lungs (grunting, fast breathing, chest wall in-drawing, oxygen saturation of <90% on room air, with crepitations on auscultation). This indeed is late onset sepsis. However, we presented nosocomial pneumonia separate from neonatal sepsis and we have included this limitation. (Page/line: 8/186-188, 20/335-337)

11. When was the Hypothermia Diagnosed at the Time of admission or during the course of admission 

Response: We considered hypothermia at the time of admission to the neonatal unit. This has been clarified. (Page/line: 8/180-181) 

 Discussion 

12. Introduction could be improved , by highlighting the purpose of the study, instead of reporting all the important significant Results in the First Paragraph

Response: We agree with the reviewer. We have added information on the purpose of the study to the first paragraph of discussion. It now reads, “This study aimed to determine the incidence and predictors of preterm neonatal mortality in a tertiary hospital in South Western Uganda. The study highlights the major predictors such as birth asphyxia, not receiving kangaroo mother care (KMC), late initiation of breastfeeding, late ANC booking and lack of ANC attendance, vaginal breech delivery, very preterm birth, respiratory distress syndrome and hypothermia at the time of admission to the neonatal unit as increasing the risk of preterm neonatal mortality”. (Page/line: 18/267-268)

13. The Respiratory Distress syndrome assessment, we would have used the SAS score , it has been used in Kiwoko Hospital 

Response: To diagnose respiratory distress syndrome, we based on WHO criteria. However, the Silverman-Andersen (SAS) score is routinely used at our institution to grade the severity of RDS before initiation of CPAP to infants with RDS.

14. When did the Hypothermia Occur, was it birth or during the admission, because hypothermia on admission to the NICU is associated with mortality. This could be added to the explanation for hypothermia 

Response: Same as # 11 above, we considered hypothermia at the time of admission. We agree with the reviewer that hypothermia at the time of admission is associated with mortality and have the information in the discussion section. (Page/line: 20/310-312)

15. The difference in the mortality could be explained by the Different levels of care in neonatology ( refer to article Survive and thrive, Inpatient care for small and sick newborn barriers By Hannah Blencowe 2015 )

Response: This has been noted and referenced appropriately. (Page/line: 19/285-288)

16. The different interventions available in Mbarara , how do they differ with other the hospitals

Response: Mbarara Regional Referral Hospital is a low resource tertiary hospital serving the South-western Uganda. The different interventions available include phototherapy machines, warmers, oxygen therapy, continuous positive airway pressure (CPAP), Kangaroo mother care. These services are lacking in the peripheral facilities. This is highlighted in the study setting section. (Page/line: 5/105-109)

17. RDS – major cause of mortality , more studies from Mulago could explain that ( Yaser et al 2012)

Response: Thank you, we have included the suggested reference. (Page/Line: 19/300-302) 

18. More explanation for Hypothermia, what prevents Hypothermia, do we have a warm chain, what interventions are available for Hypothermia apart from kangaroo mother care 

Response: We have discussed the importance of essential new born care interventions including the warm chain and timely resuscitation of neonates when indicated. (Page/ Line: 20/314-317) 

19. Need to explain, why those that needed CPAP, Died , do we have a criteria for those that have failed CPAP, if so what do we do , do we refer , we need to explain in the discussion

Response: We have acknowledged this limitation because no additional intervention was available following failed CPAP, this could have increased the contribution of RDS to mortality at our institution. (Page/line: 21/337-338)

Reviewer #2: 

Introduction:

20. This section is well written, however to increase clarity, in this sentence “It is stated that Interventions to prevent the occurrence of preterm birth are thus needed and studies focusing on this should be encouraged”, it would be useful to mention a few, since there are some of spontaneous preterm births with no cause which may not be easy to prevent.

Response: We appreciate the reviewer’s comment. We have cited examples of interventions to prevent occurrence of preterm labor. (Page/line: 3/65-66)

Methods: 

Study site:

The section is well described; my few comments and questions are as follows

21. Please add information on the ANC attendance pattern in the study site, also availability of KMC ward and size.

Response: The information on ANC attendance and KMC availability has been added. Page/line: 5/93-94 and 5/104

Participant enrollment:

22. Was the questionnaire used a newly developed by the researchers or an adopted version of questionnaire which is validated?

Response: The questionnaire was newly developed by the researchers basing on the research questions. We however pretested it before use for data collection. (Page/line: 6/131-132)

Data collection and follow up

23. What time was the baseline information collected, was it after delivery or after arrival to newborn unit, this did not come out clearly.

Response: We have clarified the point of entry into the study and collection of baseline information as after delivery. (Page/line: 6/132)

24. Authors stated that they used the first day of the last normal menstrual period (LNMP), known to be a more reliable measure of GA in a low-resource setting, however this is prone to recall bias especially if the literacy level is low, this has been a recurring problem in our unit as well, thus this can be cited as a limitation to the study. The use of Modified Ballard score could have added value.

Response: We agree with the reviewer the use of Modified Ballard could have added value. We have cited the limitation arising from using LNMP to establish gestational age. (Page/line: 20/334-335)

25. Suspected cause of death was used, why not actual cause of death since only deaths which occurred in hospital were analyzed.

Response: We used the term “suspected” cause of death because we relied on clinical cause of death since postmortem examinations were not performed for the neonates who died. (Page/line: 21/338-340)

26. For any neonates discharged alive before 28 days, post-discharge follow-up was done via a phone call at day 3, 7, 14 and 28. What was the reasons for frequent phone calls, is this part of standard of care? If not was this included in the consent form signed by parents/participants; please clarify.

Response: The phone calls at day 3, 7, 14 and 28 were not part of standard of care. The calls were aimed at keeping close contact with the study participants and to obtain the required information. This was included in the consent form signed by the parents/participants.

27. For those who died after discharge, one would have expected the deaths which occurred at home to have had verbal autopsy done to establish cause of death, this could have been possible since the deaths were confirmed by mobile phone. Was there a reason this was omitted?

Response: Verbal autopsies were carried out on phone, however, the findings were inconclusive and we categorized the deaths that occurred at home as unknown. Of the 6 neonates who died at home, 2 were reported to have died following a febrile illness, while the other 4 were reported to have been well but found dead in the bed. (Page/line: 15/248-250)

Study variables:

28. It would be very useful if the author could clearly clarify the followings in the definition of variables.

-Hypothermia: was this at admission or at any point after admission

Response: Hypothermia was at the time of admission to the neonatal unit. This has been clarified in the definition of variables. (Page/line: 8/180-181)

 -Sepsis: was this combining EOS and LOS i.e. sepsis diagnosis at any point?

Response: Yes. Sepsis accounted for any sepsis diagnosed at any point. This has been clarified in the definition of variables. (Page/line: 8/184-186)

-Hypoglycemia: at admission or any time after, also a cut-off point of 2.2 was used, which seems too low, is there an explanation? WHO refers to <2.5mmol/L. also note there is a difference between plasma and blood glucose thus it is always useful to state clearly.

Response: We used capillary blood from a heel prick for blood glucose testing using an electronic glucometer and classified hypoglycaemia as glucose <2.2mmol/l as is the routine cutoff used at the neonatal unit of Mbarara Regional Referral Hospital. This is clarified. (Page/line: 8/181-183)

-ACS: any dose at any time before delivery?

Response: Antenatal corticosteroid use was defined as having received any dose of corticosteroids at any time within 7 days before delivery. This has been clarified in the definition of variables. (Page/line: 8/173-175)

-How was RDS differentiated from congenital pneumonia? Especially for mothers with risk factors. 

Response: Thank you for raising the crucial challenge of potentially mixing up diagnoses for newborns congenital chest infection and RDS. However, when we reviewed babies at a higher risk of congenital pneumonia due to being born to women with chorioamnionitis (n=9), none had RDS.

-There is also typo instead of starting, it is written staring

Response: The typographical error has been edited. (Page/line: 7/179)

-What was the cut off for booking for ANC late? 2nd trimester third, please clarify.

Response: The cut offs for antenatal booking were defined as; first trimester booking (first antenatal visit before 13 weeks of gestation), second trimester booking (between 13 to 26 weeks of gestation) and third trimester booking (≥ 27 weeks of gestation). We considered late ANC booking as ≥13 weeks of gestation. This has been clarified in the manuscript. (Page/line: 7/175-177)

Analysis:

29. Were the interaction between variables examined in the Cox proportional hazards regression models; some of the factors seems to be correlated and can affect the outcomes i.e., a baby with sepsis may also present with hypothermia, jaundice. Also including both Birth asphyxia and Apgar score in the same model can cancel the effects. I would advise the Authors to check for correlation and interaction in the covariates included in the final models.

Response: The interaction and correlation between the variables in the Cox proportional hazards regression models were examined. There was no correlation and interaction between the variables in the multivariable model. Sepsis and jaundice were not correlated. Similarly, there was no correlation and interaction between sepsis and hypothermia at the time of admission. We however, noted correlation between birth asphyxia and Apgar score at 5 minutes, we therefore dropped Apgar score at 5 minutes from the final model. This did not alter the results as shown in table 4. (Page: 16)

Results:

30. Arrangement and labeling of tables and figures is a bit confusing. i.e what appears as table 1 in the result section I presume it should be labeled as table 3, please check.

Response: The tables are labelled as follows; table 1: sample size calculation (this is within the text), table 2: Baseline socio-demographic and obstetric characteristics, table 3: Characteristics of preterm neonates, table 4: Cumulative mortality incidence of preterm neonates, table 5: Predictors of mortality among preterm neonates born at Mbarara Regional Referral Hospital

31. For all the tables and figures please include a footnote spelling out all the abbreviations used.

Response: The footnotes have been included for all tables and figures as advised.

32. The KM figures are not clear, title shows mortality probabilities, but the Y –axis shows cumulative survival, please check and rectify.

Response: The KM curves presented depict the declining survival probabilities which mirrors the increasing mortality probabilities overtime. We thought presenting the decreasing survival on the Y-axis would be easily interpreted by the reader as commonly presented in the publications. 

33. It seems like very few-used CPAP (i.e., 216 were diagnosed with RDS but only 96 received CPAP) was it because of availability? It would be informative for policy if the median survival time was shown among those with RDS who had access to CPAP Vs those who had no access.

Response: All neonates with RDS that required CPAP during the study period received it (n=96). Majority of the neonates with RDS only required respiratory support with supplemental oxygen therapy. We acknowledge that the legend of Fig. 10, the Kaplan Meier curve for CPAP “Need for CPAP” may be misleading, it was meant to represent the neonates who required and received CPAP versus those who did not require CPAP. We have corrected the legend for Fig. 10 and presented the KM curves ventilatory support with CPAP across preterm neonates with RDS who required and received CPAP versus those who did not. Fig 10. (Page: 31)

34. Under neonatal characteristics the proportion of babies with Sepsis is not shown.

Response: This proportion is in the row number 5 of the neonatal morbidities in table 3. The portion of neonates with neonatal sepsis was 83/538 (15.4%). (Page: 13)

35. Neonatal morbidities contributing to death of preterm neonates, I am not sure if there is a clear line to differentiate NEC, nosocomial pneumonia and sepsis, it will be informative to the readers if the authors could say how they managed to do so.

Response: Differentiating necrotizing enterocolitis (NEC) and nosocomial pneumonia from sepsis may be challenging. However, basing on the 2013 WHO guidelines for diagnosis of childhood illnesses, NEC was diagnosed basing on clinical features of abdominal distension and tenderness, intolerance to feeding, bilious vomitus or fluid up the nasogastric tube, bloody stools and abdominal x-ray findings of air bubbles in the intestinal walls. This has been highlighted in the definition of variables. (Page/line: 8/189-191)

Same as in #10 above, nosocomial pneumonia was diagnosed basing on blood culture and localizing clinical features to the chest (grunting, fast breathing, chest wall in-drawing, oxygen saturation of <90% on room air, with crepitations on auscultation). This indeed is late onset sepsis. However, we presented nosocomial pneumonia separate from neonatal sepsis and we have included this limitation. (Page/line: 8/186-188, 20/335-337) 

36. It is also important to specify if sepsis was based on symptoms or cultures, since in the definition of variable it was either or

Response: Sepsis diagnosis was based on the presence of clinical features suggestive of sepsis according to the WHO’s Integrated Management of Childhood Illnesses (IMCI) algorithm and blood cultures. This has been clarified under variable definition. (Page/line: 8/184-186)

37. Preterm are prone to IVH and anemia, which contributes to increased mortality, were any of these observed in this study?

Response: It is true IVH and anemia are common occurrences in preterm neonates. These were however not observed in this study.

38. Figure 2, I would advise to use only one curve, i.e. cumulative survival curve.

Response: This is noted. The cumulative mortality curve has been dropped as advised.

39. Figure 2; the chart area will be seen more clearly if the font size for the texts was reduced.

Response: The chart is now clearer after dropping the cumulative mortality curve and formatting the figures. (Page: 14)

40. As much as results presented graphically are more visual, having a lot of KP curves around predictors is tiring. For preterm it will be more informative if the median survival time by infant demographics e.g. sex, GA (late, moderate and early preterm) or weight (LBW/SGA vs LBW/AGA)

Response: We conducted the statistical analyses for the median survival time by the different infant demographics. However the median survival time is not reached for any of the infant demographics above. We have therefore maintained the KM curves.

41. Figure 10 Need for ventilatory support with CPAP, it is obvious that those who needed CPAP may have been versus sick and thus have increased mortality. For policy implication it would be more meaningful to show if those who needed and had access vs those who needed but did not have an access

Response: As in #33 above, all neonates with RDS that required CPAP during the study period received it (n=96). Majority of the neonates with RDS only required respiratory support with supplemental oxygen therapy. We acknowledge the miss representation of the KM curve for CPAP in Fig. 10 as “Need for CPAP”, it was meant to represent the neonates who required and received CPAP versus those who did not require CPAP. We have corrected the legend for Fig. 10 and presented the KM curves for ventilatory support with CPAP across preterm neonates with RDS who received CPAP versus those who did not. (Page: 31)

42. I suggest table 4 be omitted, since What is presented in table 4 is a duplicate of what is presented in figure 2,

Response: Table 4 has been omitted as suggested. 

Predictors of mortality among preterm neonates;

43. For strong predictors it will be useful to focus more on those with high aHR and significant CI. Eg Birth asphyxia (aHR 11.9), not receiving KMC (aHR 9.14), late initiation of BF (aHR 8.56), based on these results suggests investment in essential newborn care could reduce mortality in these babies

Response: This has been noted. We have focused more on the strong predictors with high aHR as recommended. Essential newborn care has been emphasized in the discussion as recommended. (Page/line: 15/256-263)

44. Some of the confidence intervals are very wide i.e. birth asphyxia (aHR, 11.90; 95% CI: 4.08-34.70). Can the authors comment on this and the implication it has on their results and recommendations.

Response: Some of the confidence intervals are wide (for example for birth asphyxia and late initiation of breastfeeding) which implied loss of precision due to small numbers of participants with birth asphyxia and those with early initiation of breastfeeding. However we believe that our findings are informative because the lower boundaries of these confidence intervals for each of the point estimates are above two-fold. 

Discussion:

45. It would have been more informative if the author could discuss in relation to feasible interventions to overcome the problems they identified i.e. based on the strong predictors enhancement of ENC should be more discussed, I guess many investors would give the money to improve this over postnatal surfactant, which has less value if ENC is not adequate.

Response: We agree with the reviewer and have included this information in the discussion section. (Page/line: 20/314-317)

46. References: 19, 37, 40 and 42 need to be appropriately cited, they should clearly indicate the source; if they are from the internet then URL and date retrieved should be included.

Response: The references 19, 37, 40 and 42 have been appropriately cited and the source has been clearly indicated.

---

## [Decision Letter · Decision Letter 1]

11 Oct 2021

PONE-D-21-19132R1Incidence and predictors of preterm neonatal mortality at Mbarara Regional Referral Hospital in South Western UgandaPLOS ONE

Dear Dr. Tibaijuka,

Thank you for submitting your manuscript to PLOS ONE. After careful consideration, we feel that it has merit but does not fully meet PLOS ONE’s publication criteria as it currently stands. Therefore, we invite you to submit a MINOR revised version of the manuscript that addresses the points raised during the review process.

Thank you for submitting revised manuscript. You have addressed most of comments raised by the reviewers. Please could you address the concerns from the reviewer 1 and re-submit.

We look forward to receiving your revised manuscript.

Kind regards,

Sajid Bashir Soofi

Academic Editor

PLOS ONE

Journal Requirements:

Additional Editor Comments (if provided):

Thank you for submitting your revised manuscript to PLOS ONE. But reviewer has some minor comments, please submit a revised version of the manuscript that addresses the points raised during the review process.

Reviewers' comments:

Reviewer's Responses to Questions

**Comments to the Author**

1. If the authors have adequately addressed your comments raised in a previous round of review and you feel that this manuscript is now acceptable for publication, you may indicate that here to bypass the “Comments to the Author” section, enter your conflict of interest statement in the “Confidential to Editor” section, and submit your "Accept" recommendation.

Reviewer #2: (No Response)

Reviewer #3: (No Response)

2. Is the manuscript technically sound, and do the data support the conclusions?

Reviewer #2: Yes

Reviewer #3: Yes

3. Has the statistical analysis been performed appropriately and rigorously? 

Reviewer #2: Yes

Reviewer #3: Yes

4. Have the authors made all data underlying the findings in their manuscript fully available?

Reviewer #2: Yes

Reviewer #3: (No Response)

5. Is the manuscript presented in an intelligible fashion and written in standard English?

Reviewer #2: Yes

Reviewer #3: Yes

6. Review Comments to the Author

Reviewer #2: Most of the comments have been sufficiently addressed except the following

1. Labeling of the tables reviewer responded that "The tables are labelled as follows; table 1: sample size calculation (this is within the text), table 2: Baseline socio-demographic and obstetric characteristics, table 3: Characteristics of preterm neonates, table 4: Cumulative mortality incidence of preterm neonates, table 5: Predictors of mortality among preterm neonates born at Mbarara Regional Referral Hospital .

however in the document table section they are labeled as table 2, table 1, table 2 then table 5, please correct.

2. Wide confidence interval as this could results in loss of precision should at least be mentioned to caution the readers.

3. Based on the objective of the study "to determine the incidence and predictors of preterm

neonatal mortality" i am not convinced that KM curves for each risk factor is necessary especially since they are not adjusted. Multvariate cox regression answer the question well. KM curves could be attached as supplementary material if needed.

Reviewer #3: well articulated manuscript on an important public health subject especially in the settings where the study is carried out .

the paper highlights important interventions required to improve preterm outcomes.

recommend it for publication and dissemination

7. PLOS authors have the option to publish the peer review history of their article (what does this mean?). If published, this will include your full peer review and any attached files.

Reviewer #2: No

Reviewer #3: **Yes: **Shabina Ariff

---

## [Author Response · Author response to Decision Letter 1]

11 Oct 2021

RESPONSE TO REVIEWERS’ COMMENTS

Reviewer #2: Most of the comments have been sufficiently addressed except the following

1. Labelling of the tables reviewer responded that "The tables are labelled as follows; table 1: sample size calculation (this is within the text), table 2: Baseline socio-demographic and obstetric characteristics, table 3: Characteristics of preterm neonates, table 4: Cumulative mortality incidence of preterm neonates, table 5: Predictors of mortality among preterm neonates born at Mbarara Regional Referral Hospital. However in the document table section they are labelled as table 2, table 1, table 2 then table 5, please correct.

Response: This is noted and has been corrected. The tables are now labelled as follows; table 1: Sample size calculation, table 2: Baseline socio-demographic and obstetric characteristics, table 3: Characteristics of preterm neonates, table 4: Predictors of mortality among preterm neonates born at Mbarara Regional Referral Hospital.

2. Wide confidence interval as this could results in loss of precision should at least be mentioned to caution the readers.

Response: We acknowledge this limitation and have stated it as such. (Page/Line: 21/338-340)

3. Based on the objective of the study "to determine the incidence and predictors of preterm neonatal mortality", I am not convinced that KM curves for each risk factor is necessary especially since they are not adjusted. Multivariate cox regression answer the question well. KM curves could be attached as supplementary material if needed.

Response: We acknowledge that multivariable cox regression models answer the objective of this study and that KM curves are not adjusted. We have therefore attached the KM curves as supplementary material as advised.

Reviewer #3: Well articulated manuscript on an important public health subject especially in the settings where the study is carried out. The paper highlights important interventions required to improve preterm outcomes.

Recommend it for publication and dissemination.

Response: We appreciate the reviewer for the positive comments.

---

## [Editor Report · Decision Letter 2]

18 Oct 2021

Incidence and predictors of preterm neonatal mortality at Mbarara Regional Referral Hospital in South Western Uganda

PONE-D-21-19132R2

Dear Dr. Leevan Tibaijuka

We’re pleased to inform you that your manuscript has been judged scientifically suitable for publication and will be formally accepted for publication once it meets all outstanding technical requirements.

Kind regards,

Prof Sajid Bashir Soofi

Academic Editor

PLOS ONE
---

## [Editor Report · Acceptance letter]

20 Oct 2021

PONE-D-21-19132R2 

Incidence and predictors of preterm neonatal mortality at Mbarara Regional Referral Hospital in South Western Uganda 

Dear Dr. Tibaijuka:

I'm pleased to inform you that your manuscript has been deemed suitable for publication in PLOS ONE. Congratulations! Your manuscript is now with our production department. 

Kind regards, 

on behalf of

Professor Sajid Bashir Soofi 

Academic Editor

PLOS ONE